# Identifying priority challenges and solutions for COVID-19 vaccine delivery in low- and middle-income countries: A modified Delphi study

**Archchun Ariyarajah[1], Isha Berry[1], Victoria Haldane[1], Miranda Loutet[1], Fabio Salamanca-Buentello[2], Ross E. G. Upshur[1,2] ***

1 Dalla Lana School of Public Health, University of Toronto, Toronto, ON, Canada, 2 Bridgepoint Collaboratory for Research and Innovation, Lunenfeld-Tanenbaum Research Institute, Sinai Health, Toronto, ON, Canada

☯ These authors contributed equally to this work.

* ross.upshur@utoronto.ca

**Data Availability Statement:** The de-identified dataset is available as a supplementary file. This dataset includes all data sociodemographic data

## Abstract

### Background

The rapid implementation of global COVID-19 vaccination programs has surfaced many challenges and inequities, particularly in low- and middle-income countries (LMICs). However, there continues to be a lack of consensus on which challenges are global priorities for action, and how to best respond to them. This study uses consensus-based methods to identify and rank the most important challenges and solutions for implementation of COVID-19 vaccination programs in LMICs.

### Methods

We conducted a three-round modified Delphi study with a global panel of vaccine delivery experts. In Round I, panelists identified broad topical challenges and solutions. Responses were collated and coded into distinct items. Through two further rounds of structured, iterative surveys panelists reviewed and ranked the identified items. Responses were analyzed qualitatively and quantitatively to achieve consensus on the most important COVID-19 vaccine delivery challenges and solutions.

### Results

Of the 426 invited panelists, 96 completed Round I, 56 completed Round II, and 39 completed Round III. Across all three rounds there was equal representation by gender, and panelists reported work experience in all World Bank regions and across a variety of content areas and organizations. Of the 64 initially identified items, the panel achieved consensus on three challenges and 10 solutions. Challenges fell under themes of *structural factors* and *infrastructure and human and material resources*, while solutions also included items within themes of *communication, community engagement, and access* and *planning, processes, and operations.*

from Round 1, as well as ranking responses from Rounds 2 and 3. The free-text response data from Round 1 cannot be made publicly available due to risk of panelist identification.

**Funding:** This work received funding from the University of Toronto Implementation Science Cluster Trainee Program and the University of Toronto Student Engagement Award. Funders had no role in the design and conduct of the study; collection, management, analysis, and interpretation of the data; preparation, review, or approval of the manuscript; or decision to submit the manuscript for publication.

**Competing interests:** Authors declare no competing interests.

## Conclusion

COVID-19 vaccine delivery is challenged by long-standing and structural inequities that disadvantage health service delivery in LMICs. These findings can, and should, be used by global health organizations to efficiently and optimally direct resources to respond to these key challenges and solutions.

## Introduction

Effective vaccines and robust vaccination programs are crucial components of the COVID-19 pandemic response. Multiple effective COVID-19 vaccines were developed and approved for emergency use authorization in less than a year due to at-risk investments from governments and new vaccine platforms such as those based on mRNA [1]. The rapid development and delivery of COVID-19 vaccines required significant global coordination, which led to the creation of COVAX—the vaccines pillar of the Access to COVID-19 Tools (ACT) Accelerator. COVAX is a multilateral initiative co-led by Gavi, the Coalition for Epidemic Preparedness Innovations (CEPI), and the World Health Organization (WHO) to accelerate the development, production, and equitable access to COVID-19 vaccines [2]. COVAX pools the resources of participating countries to speed up the development and manufacturing of vaccines, with investment risks shared across all participating countries [3]. To ensure vaccine equity amongst participating COVAX countries, self-financing countries pay for vaccine doses, while these are provided at-cost or freely to countries that cannot afford them [3].

Although COVAX's funding mechanisms to speed up vaccine development and production have been viewed as a success, COVAX's ability to ensure global vaccine equity has been questioned [4]. The rapid implementation of global COVID-19 vaccination programs has faced many challenges and inequities, particularly in low- and middle-income countries (LMICs). As of February 2022, the global COVID-19 vaccination rate is 133.5 doses administered per 100 people [5]. However, regional inequities persist, with lower-income countries reporting a rate of 17.0 per 100 people, and high-income countries reporting 184.1 per 100 people [5]. The Regional Office for Africa (AFRO) is the WHO region with the lowest vaccination rate, with a rate of 21.7 per 100 people [5].

Such global discrepancies are multifactorial. Issues related to limited vaccine access in LMICs include: lack of global coordination for equitable access of vaccines [6]; vaccine nationalism, where high-income countries prioritize their own public health needs above global needs [4, 7]; bilateral agreements for vaccines between governments of high-income countries and manufacturers [6]; limited funding of COVAX [8]; lack of transparency of agreements and prices [4]; and limited manufacturing capacity in LMICs due to intellectual property rights and limited technical capacity [6, 9]. Challenges have also arisen during on-the-ground implementation of COVID-19 vaccination programs in LMICs, such as in-country regulatory processes, logistical constraints, inequitable distribution, and vaccine hesitancy [6, 10].

Although similar concerns have previously been identified during other pandemics such as the 2009 H1N1 influenza pandemic, there continues to be a lack of consensus on which are the most pressing challenges, particularity in the context of LMICs [11]. A better understanding of these challenges may also offer robust opportunities and lessons learned to identify potential solutions. Given the limited funding and resources in LMICs to implement new vaccination programs, a consolidated ranking of challenges and solutions for vaccine delivery can be used to optimize resource allocation and galvanize local response efforts for the COVID-19

response. With this goal in mind, we conducted a three-round modified Delphi study to build consensus on the most important challenges and solutions for implementation of COVID-19 vaccination programs in LMICs.

## Materials and methods

The Delphi process uses a series of sequential questionnaires to collect and distil knowledge from a panel of experts, who are anonymous to each other, to build reliable group consensus [12]. In line with previous studies, we determined that three Delphi rounds using online structured feedback would be sufficient to achieve consensus and stability within our expert panel [13].

### Panelists & recruitment

To compose our panel, we identified participants with expertise in global vaccine delivery and implementation through grey and peer-reviewed published literature and invited them to participate in the Delphi study. Individuals were eligible if they were ≥18 years of age, had access to the internet, could read and write in English, and had been identified as experts in vaccine delivery and implementation. We purposely sought representativeness with respect to gender, geographic distribution, and content area expertise.

Panelists were recruited using purposive and criterion sampling methods. The initial list of experts was identified through professional contacts of the study team, and by way of scanning both grey and peer-reviewed published literature for authors with evidence of vaccine delivery expertise (**S1 Table**). We also identified expert members of publicly listed immunization committees and of panels from various global health organizations, as well as authors of global immunization reports (**S1 Table**). Panelists were invited to participate through standardized emails that were sent to their publicly reported email addresses. We also utilized snowball sampling to identify additional experts who may not have been captured in the literature. Upon invitation into the study, panelists were offered the opportunity to nominate up to three other experts within their network. Each new nominated panelist was subsequently invited into the study, and similarly offered the opportunity to nominate additional experts. This process was repeated until saturation was reached, that is, when newly nominated individuals suggested already-invited experts. Given that some panelists may have multiple organizational affiliations, and therefore could have multiple email addresses, the study team manually reviewed the list of nominated panelists to remove duplicates.

Throughout the Delphi process, panelists were blinded to each other's identity, except for the panelists who provided referrals. Survey content was never associated with a panelist identifier, and only the study team could associate panelists with responses.

### Data collection & analysis

Data were collected over three rounds: R-I (April 20, 2021-May 18, 2021), R-II (August 16, 2021- September 11, 2021), and R-III (October 5, 2021- October 30, 2021). For each round, data were collected using the Research Electronic Data Capture (REDCap) survey platform hosted at the University of Toronto [14]. Panelists with incomplete responses were sent weekly reminders until the round closed to increase response rates.

R-I was designed to elicit broad and general concepts from the panelists using unstructured, open-ended questions. Panelists were asked to identify three to five of the most important challenges in i) distribution, ii) prioritization, and iii) administration of COVID-19 vaccines in LMICs, along with three to five potential solutions to these challenges. To ensure consistency, we provided standardized definitions for vaccine distribution, prioritization, and

administration at the start of the survey (**Table 1**). A minimal set of socio-demographic data were also collected, including gender, geographic location of expertise (select multiple option), content areas of expertise (select multiple option), organizational affiliation, and years of experience.

Data from R-I were combined, and members of the research team (AA, IB, VH, ML) synthesized the challenges and solutions into a taxonomy according to themes and topics. Our analysis was guided by a modified COVID-19 Vaccine Introduction Readiness Assessment Tool (VIRAT/VRAF 2.0) framework, which draws on a set of key country indicators for vaccine deployment readiness as developed by the WHO and World Bank [15]. The coded themes and topics were synthesized into 64 mutually exclusive items. In R-II, the compiled list of challenges and solutions was sent to all panelists who completed R-I. Panelists were asked to rank the importance of each item using a five-point Likert scale (not at all important, slightly important, moderately important, very important, extremely important).

Analysis of scores from R-II was conducted to eliminate challenges and solutions that did not reach an *a priori* cut-off of 75% agreement (defined as responding 'very important' or 'extremely important'). In R-III, the remaining shortlisted challenges and solutions were sent back to panelists who completed R-II. Panelists were asked to provide their level of agreement with each item being one of the most important challenges/solutions. Agreement was measured using a five-point Likert scale (strongly disagree, disagree, neutral, agree, strongly agree). Panelists reporting 'strongly disagree' or 'disagree' for any item were also offered a free-text response option to provide dissenting views.

Data from R-III were analyzed quantitatively and qualitatively to determine consensus using two prespecified criteria. First, the degree of consensus for each item was graded following previously established methods [16]: 100% agreement was graded as 'U' (unanimous); 90%-99% agreement was 'A'; 80%-89% agreement was 'B', and items with less than 80% agreement were graded as 'NC' (no consensus). Agreement was defined as responding 'agree' or 'strongly agree'. Second, comments provided by dissenting respondents were reviewed qualitatively. Any items with dissenting views could only achieve consensus if none of the dissenting views were fundamentally incompatible with the inclusion of that item. This approach recognizes that essential insights can be tendered by a minority of decision-makers and attends to the substance of minority opinions [13].

Demographic data from each round were tabulated to examine panel representativeness over time. All qualitative analyses were conducted in QSR NVivo 12 (QSR International), and quantitative analyses were conducted in Stata 16.0 (StataCorp, College Station, TX, USA).

## Ethics

This study received ethical approval from the University of Toronto Office of Research Ethics (Protocol Number: 40797). All participants provided online written informed consent before being permitted to access survey questions.

**Table 1. Standard definitions on vaccine distribution, prioritization, and administration provided to Delphi participants during Round I.**

| Category | Definition |
| --- | --- |
| **Distribution** | The acquisition, storage, and deployment of COVID-19 vaccines. |
| **Prioritization** | The order in which COVID-19 vaccines should be distributed to populations and geographical areas, and for special considerations for program implementation. |
| **Administration** | The actions and mechanisms that ensure COVID-19 vaccines are provided to the population safely and effectively. |

## Results

In this section we first summarize the characteristics of panelists who participated in each of the three rounds of the Delphi study. We then describe the taxonomy of challenges and solutions synthesized from the free text responses in R-1, followed by the top ranked COVID-19 vaccine implementation challenges and solutions from R-II and those that reached consensus in R-III.

### Panelists

We invited 426 vaccine implementation experts to our Delphi study; panelist response and retention rates varied over time. In R-I, 22% (96/426) of invited panelists completed the survey, in R-II 58% (56/96) of the remaining panelists participated, and in R-III 70% (39/56) participated (**Fig 1**). Across all three rounds there was balanced representation by gender, and panelists reported work experience in all World Bank regions and across a variety of content areas and organizations (**Table 2**). The highest proportion of panelists reported work experience in Sub-Saharan Africa (R-I: 41.7%, R-II: 39.3%, R-III: 38.5%), Latin America and the Caribbean (R-I: 38.5%, R-II: 44.6%, R-III: 46.2%), and South Asia (R-I: 35.4%, R-II: 30.4%, R-III: 28.2%). Participation was also initially high amongst those with experience in East Asia and the Pacific but dropped in subsequent rounds (R-I: 26.0%, R-II: 17.9%, R-III: 18.0%). The majority of panelists reported public health and surveillance expertise (R-I: 63.5%, R-II: 67.9%, R-III: 69.2%) and organizational affiliations with a multilateral (R-I: 37.5%, R-II: 37.5%, R-III: 38.5%) or research (R-I: 25.0%, R-II: 32.1%, R-III: 30.8%) organization. Most respondents had ≥10 years of experience (R-I: 79.2%, R-II: 87.5%, R-III 84.6%).

### Challenges and solutions

In R-I, many respondents identified distribution and administration challenges relating to vaccine access, cold chain capacity, as well as infrastructure and human resource capacity; in terms of prioritization challenges panelists overwhelmingly identified inequity. Solutions proposed by many respondents included stronger coordination at and across all levels of government, and development of effective communication plans with social mobilization and community engagement. While responses were provided under the domains of distribution, prioritization, and administration, there were substantial overlaps across these categories. Through our qualitative analysis 36 challenges and 28 solutions were identified, and these were categorized into four cross-cutting themes: infrastructure and human and material resources; planning, processes, and operations; communication, community engagement and access; and structural factors (**S2 Table**). Most challenges were related to structural factors (n = 11) and planning, processes, and operations (n = 10), while most solutions were related to infrastructure and human and material resources (n = 9).

In R-II, seven challenges and 10 solutions met the cut-off ranking for importance, including at least one element in each of the cross-cutting themes (**S2 Table**). The highest ranked challenges were *inequitable ability of LMICs to acquire vaccines due to limited independent purchasing power and dependence on COVAX* (87.5% ranked as extremely/very important) and *inadequate cold chain and storage infrastructure* (85.7%). The highest ranked solutions were to *collaborate with local traditional, civil, and religious leaders to address concerns* (90.0%) and *strengthening cold chain capacity* (83.3%).

In R-III, three challenges and 10 solutions achieved consensus, though consensus grades varied (**Tables 3 and S2**). The most important challenges for implementation of COVID-19 vaccination programs in LMICs were *insufficient health system capacity for routine care and COVID-19 care* (94.9% responded agree/strongly agree) and *insufficient operational funding*

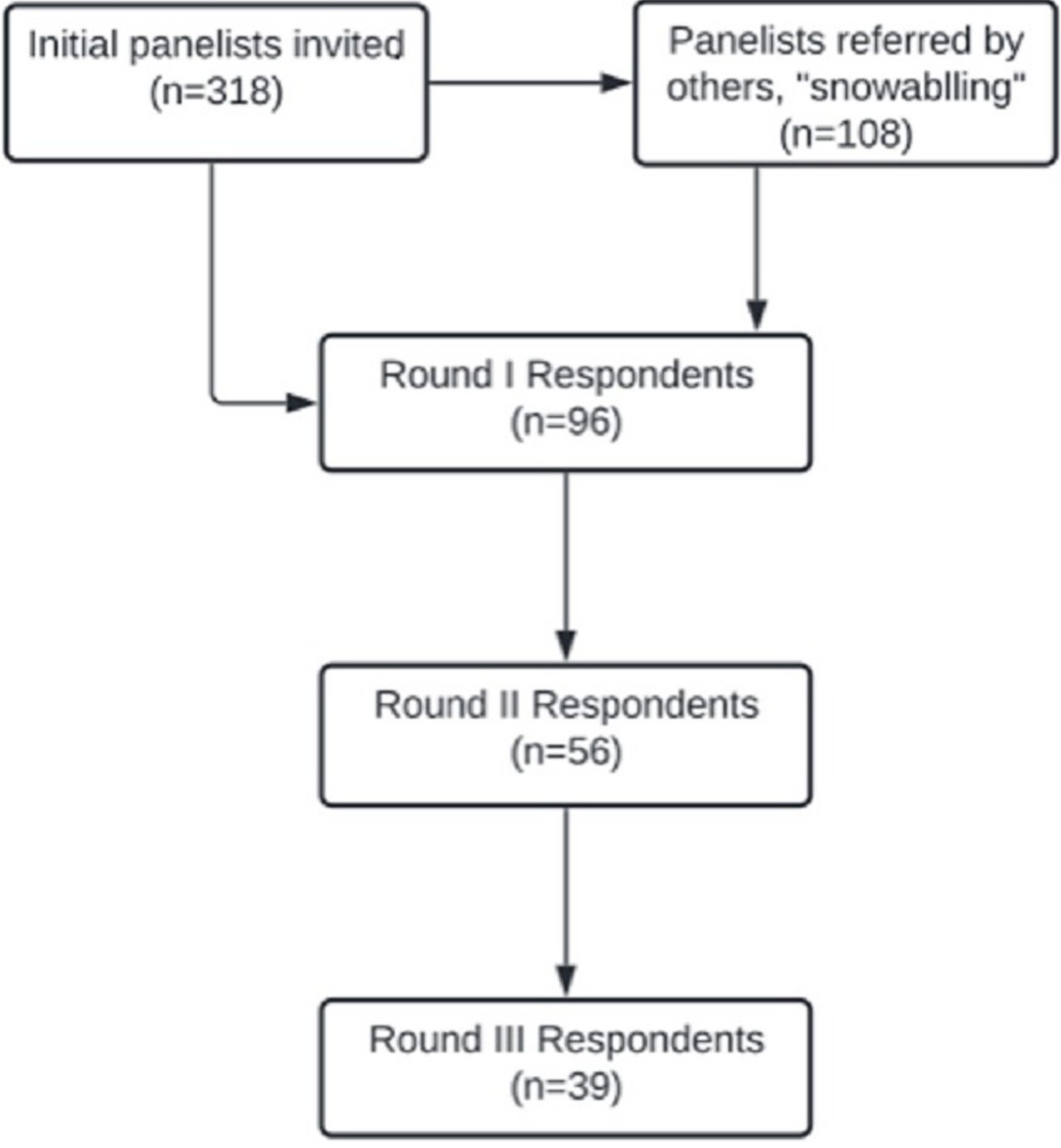

**Fig 1. Flow chart of panelist recruitment and retention.**

*within countries* (92.3%). The category *inadequate cold chain and storage infrastructure* was also identified but received a lower ranking (86.8%). Panelists with dissenting views indicated that disagreement was due to local context and the level at which these challenges were occurring, but no fundamentally incompatible reasons were identified. All 10 solutions achieved consensus, with higher ranking solutions (>90%) related to themes of infrastructure and human and material resources; planning, processes, and operation; and communication,

**Table 2. Demographic characteristics of Delphi panelists in Round I, Round II, and Round III.**

| | | Round I | Round II | Round III |
|---|---|---|---|---|
| | | N (%) | N (%) | N (%) |
| **Total Respondents** | | 96 | 56 | 39 |
| **Gender[1]** | | | | |
| | Man | 50 (52.1) | 29 (51.8) | 21 (53.9) |
| | Woman | 43 (44.8) | 25 (44.6) | 17 (44.6) |
| | Prefer Not to Disclose | 1 (1.0) | 1 (1.8) | 0 (0.0) |
| **Region of Work Experience[2]** | | | | |
| | East Asia & Pacific | 25 (26.0) | 10 (17.9) | 7 (18.0) |
| | Europe & Central Asia | 23 (24.0) | 13 (23.2) | 9 (23.1) |
| | Latin America & Caribbean | 37 (38.5) | 25 (44.6) | 18 (46.2) |
| | Middle East & North Africa | 23 (24.0) | 12 (21.4) | 8 (20.5) |
| | North America | 17 (17.7) | 11 (19.6) | 7 (18.0) |
| | South Asia | 34 (35.4) | 17 (30.4) | 11 (28.2) |
| | Sub-Saharan Africa | 40 (41.7) | 22 (39.3) | 15 (38.5) |
| | Prefer Not to Disclose | 1 (1.0) | 0 (0.0) | 0 (0.0) |
| **Affiliated Organization[1]** | | | | |
| | Funding Agency/Donor | 4 (4.2) | 2 (3.6) | 0 (0.0) |
| | Government (including governmental public health organization) | 8 (8.3) | 2 (3.6) | 2 (5.1) |
| | Health Care Facility | 3 (3.1) | 1 (1.8) | 1 (2.6) |
| | Industry (e.g., pharmaceutical company, corporation, etc.) | 3 (3.1) | 1 (1.8) | 0 (0.0) |
| | Research organization/ Academic institution (e.g., university, college, etc.) | 24 (25.0) | 18 (32.1) | 12 (30.8) |
| | Non-governmental Organization | 10 (10.4) | 7 (12.5) | 5 (12.8) |
| | Multilateral Organization (e.g., WHO, World Bank) | 36 (37.5) | 21 (37.5) | 15 (38.5) |
| | Public-private Partnership (e.g. GAVI) | 3 (3.1) | 1 (1.8) | 1 (2.6) |
| | Independent Consultant | 3 (3.1) | 2 (3.6) | 2 (5.1) |
| **Content Area Expertise[2]** | | | | |
| | Clinical Practice | 12 (12.5) | 7 (12.5) | 5 (12.8) |
| | Population Health Research | 28 (29.2) | 19 (33.9) | 14 (35.9) |
| | Lab Research | 9 (9.4) | 4 (7.1) | 2 (5.1) |
| | Vaccine Development | 16 (16.7) | 12 (21.4) | 5 (12.8) |
| | Policy-making & Governance | 41 (42.7) | 23 (41.1) | 18 (46.2) |
| | Public Health & Surveillance | 61 (63.5) | 38 (67.9) | 27 (69.2) |
| | Logistics & Supply Chains | 23 (24.0) | 11 (19.6) | 9 (23.1) |
| | Leadership & Management | 35 (36.5) | 19 (33.9) | 14 (35.9) |
| | Program Development & Evaluation | 32 (33.3) | 22 (39.3) | 18 (46.2) |
| | Industry | 3 (3.1) | 1 (1.8) | 0 (0.0) |
| **Years of Experience** | | | | |
| | <5 years | 1 (1.0) | 1 (1.8) | 1 (2.6) |
| | 5–9 years | 17 (17.7) | 5 (8.9) | 4 (10.3) |
| | 10+ years | 76 (79.2) | 49 (87.5) | 33 (84.6) |

**Note**: [1]Missing data for gender n = 2 (2.1%), organization n = 2 (2.1%), and years of experience n = 2 (2.1%).
[2]Values will not sum to 100% because participants could select all that apply.

community engagement, and access. There was unanimous agreement on the importance of *developing flexible national plans* (100%). Other highly ranked solutions included to *develop collaborative communication strategies to address misinformation, disinformation, and vaccine*

**Table 3. Ranked consensus statements on the most important challenges and solutions for implementation of COVID-19 vaccination programs in low- and middle-income countries (n = 39).**

| Theme | Challenge | Consensus Ranking |
|---|---|---|
| **Infrastructure, and human and material resources** | Insufficient health system capacity to simultaneously deliver routine primary care and COVID-19 vaccines at the required scale and speed. | A (94.9) |
| **Structural factors** | Insufficient operational funding within countries to create infrastructure and mobilize human resources for vaccine distribution and administration. | A (92.3) |
| **Infrastructure, and human and material resources** | Inadequate cold chain and storage infrastructure, including insufficient and insecure facilities, unreliable power supply, and absent or poorly maintained equipment. | B (86.8) |
| **Theme** | **Solution** | |
| **Planning, processes, and operations** | Develop flexible plans at the national level in anticipation of multiple scenarios to ensure effective response to changing situations such as supply availability, public perceptions, and the epidemiological situation. | U (100.0) |
| **Communication, community engagement, and access** | Develop communication strategies through the collaboration of multiple partners (e.g., academia, public health agencies, regulators, media) to counter misinformation, disinformation, and vaccine hesitancy. | A (94.9) |
| **Communication, community engagement, and access** | Collaborate with local traditional, civil, and religious leaders to address concerns such as vaccine hesitancy. | A (94.5) |
| **Infrastructure, and human and material resources** | Provide dedicated training for health care workers focusing on interpersonal communication skills that can facilitate addressing concerns and doubts about the vaccines and the vaccination programmes. | A (92.3) |
| **Infrastructure, and human and material resources** | Strengthen cold chain capacity through improved transportation, enhanced storage space, temperature monitoring, etc. | A (92.3) |
| **Planning, processes, and operations** | Develop centralized surveillance systems and digital tools that allow for continuous monitoring and evaluation of key vaccine indicators such as doses distributed, vaccine coverage, adverse events following immunization (AEFIs). | A (92.1) |
| **Communication, community engagement, and access** | Design strategic, context-sensitive risk communication materials and awareness campaigns tailored to different communities, including campaigns targeted at health care workers. | B (89.5) |
| **Communication, community engagement, and access** | Prioritize the vaccination of populations involved in the maintenance of essential services such as health care, education, and food industry workers. | B (87.2) |
| **Structural factors** | Create a new or reformed global mechanism and binding agreement, with clear accountability, to better regulate and ensure equitable access to, and supply of, vaccines to LMICs regardless of their purchasing power. | B (82.1) |
| **Infrastructure, and human and material resources** | Deploy mobile units to vaccinate remote or hard-to-reach populations. | B (81.1) |

**Note**: LMICs, low- and middle-income countries

Grade Unanimous (U) is 100% agreement, A is 90–99% agreement, B is 80–89% agreement. We defined consensus as agreement among more than 80% of the panelists.

*hesitancy* (94.9%); *collaborate with local traditional, civil, and religious leaders to address concerns* (94.5%), *health care worker vaccine training* (92.3%), *strengthening cold chain capacity* (92.3%), and *develop centralized surveillance systems* (92.1%). There was lower agreement on solutions to *design tailored awareness campaigns (e.g., for vaccine access and registration)* (89.5%), *prioritizing vaccination of essential workers* (87.2%), *creating a new or reformed global mechanism to ensure vaccine equity* (82.1%), and *using mobile units to vaccinate hard-to-reach populations* (81.1%). There were no fundamentally incompatible views identified amongst those in disagreement with the solutions.

## Discussion

Drawing on a three-round modified Delphi study conducted amongst a panel of global vaccine delivery and implementation experts, we identified and ranked key challenges to COVID-19 vaccine-roll out faced in LMICs, as well as potential solutions to address them. Our findings underscore the ways in which COVID-19 vaccine delivery is challenged by long-standing and

structural inequities that disadvantage health service delivery in LMICs. The COVID-19 pandemic has also amplified global inequities to create new and pressing barriers to vaccine delivery and implementation. Despite complex challenges, panelists offered several innovative and cross-cutting solutions to strengthen not only COVID-19 vaccination programs in LMICs but also health systems and health service delivery. Importantly, solutions emphasized the need for concrete steps towards global vaccine equity.

The COVID-19 pandemic has placed unprecedented demands on health systems globally, particularly in LMICs [17]. Ongoing waves of infections, the emergence of new variants, and limited vaccination, continue to strain already weakened health systems and amplify existing systems challenges. Yet, these same systems, which are at the forefront of providing care for people with COVID-19, must now simultaneously design and implement a mass COVID-19 vaccination campaign aimed at delivering multiple doses and vaccine types. Our findings highlight the ubiquitous challenge of insufficient health system capacity to deliver both routine care and COVID-19 vaccines at the required scale and speed. Our panel also emphasized how vaccine efforts were stymied by inadequate infrastructure, limited human resources, and a lack of operational funding to immediately address these gaps. Past mass vaccination efforts in LMICs have similarly been thwarted by the interconnected health systems challenges of fragmentation, deficient health workforce capacity, and lack of investment [18]. Indeed, persistent structural challenges have long slowed progress towards global health goals to protect communities against key vaccine-preventable diseases that disproportionately impact people living in LMICs. For example, evidence on the impact of accelerated measles elimination activities in several LMICs found that countries with relatively stronger health systems benefitted from ambitious vaccination programs, whereas those with weaker health systems faced greater staff workloads and service interruptions while implementing mass vaccination campaigns [19].

Addressing these challenges to provide effective, sustainable, and comprehensive health services, including mass vaccination programs, demands solutions that improve vaccination program delivery while also reimagining broader health systems and global health equity. Our panelists identified several solutions to improve delivery and implementation of COVID-19 vaccine programs. Chief amongst these is the development of flexible national plans to guide vaccine program roll-out. Other vaccine preventable diseases have similarly benefited from global and national strategies to guide program implementation and ensure comprehensive access. For example, at the global level, the Global Polio Eradication Initiative, launched following the 1988 World Health Assembly, has aligned partners and national governments under a polio eradication strategy, including supporting the development of contextually relevant national plans and implementation strategies [20]. These efforts have led to the elimination of all but 0.1% of global polio cases [20].

However, the development of plans alone does not ensure programmatic success. Ensuring vaccination against polio and other vaccine preventable diseases has long required robust, community-engaged, and tailored communication strategies [21]. Comprehensive communication is crucial in mitigating vaccine hesitancy and meeting people's information needs, especially in underserved or otherwise marginalized communities. Our panel underscored that COVID-19 demands similarly robust communication efforts. Panelists highlighted the importance of collaboration with multiple partners including community leaders, whose involvement has been essential for community acceptance of vaccines prior to the pandemic [22, 23]. Importantly, panelists called attention to the role of health workers as crucial messengers of vaccine information and recommended health worker training focusing on interpersonal communication skills to address concerns and doubts about vaccines and vaccination programs. Studies have found that health worker attitudes towards vaccines impact patient trust and vaccine uptake across vaccine preventable diseases [24]. Notably, though vaccine hesitancy

was identified as a challenge in early rounds of our Delphi, it did not meet consensus. There is a growing body of evidence reporting that COVID-19 vaccine acceptance in LMICs is relatively high, and tailored information provided by health workers can be used to address specific concerns [25].

Our panelists also offered solutions to strengthen health systems and improve health service delivery in the long-term, while ensuring effective COVID-19 vaccine roll-out. For example, panelists noted the importance of developing centralized surveillance systems and digital tools allowing for continuous monitoring and evaluation of key indicators and adverse events following immunization (AEFIs). Strengthening COVID-19 vaccine monitoring and evaluation both benefits COVID-19 vaccination efforts and builds health systems capacity. Investing in and expanding surveillance, monitoring, and evaluation capacities can support other vaccination and disease programs that leverage these systems and the skilled workforce that maintains them. For example, networks established for polio, measles, and rubella surveillance have been leveraged to support programs for other vaccine preventable diseases in LMICs, including, among others, neonatal tetanus, yellow fever, cholera, and meningitis [26]. Such mutual benefits are also seen in improvements to health service delivery that can extend beyond use in vaccine delivery. Panelists, for instance, emphasized the need to invest in infrastructure, notably cold chain capacity, and engage community-centered models of care, such as mobile units to vaccinate remote or hard-to-reach communities. However, surveillance systems, infrastructure, and mobile models of care must be matched with sustainable investment to ensure that interventions strengthen the health system both during and after COVID-19. Indeed, these solutions are not COVID-19 specific and their emphasis in our findings ultimately point to the vast global inequities that both weaken health systems and prevent us from achieving global vaccination goals and universal health coverage.

Finally, COVID-19 vaccine delivery and implementation has thus far offered a tragic example of the limits of global solidarity. Despite attempts to ensure equitable global access to vaccines through COVAX, none of the targets set in the Strategy to Achieve Global COVID-19 Vaccination by Mid-2022 have been met [27]. Indeed, in early 2022 much of the world remains un- or under-vaccinated, with inequities between LMICs and high-income countries widening in vaccine coverage for first, second, and third doses [28]. Panelists proposed the creation of a new or reformed global mechanism and binding agreement, with clear accountability, to better regulate and ensure equitable access to, and supply of, vaccines to LMICs regardless of their purchasing power. Considering the stark differences in vaccination rates globally, questions have been raised about the ability of COVAX to deliver on its mandate given limited incentives for global collaboration, rising vaccine nationalism, and long-standing global power imbalances disadvantaging LMICs [29, 30].

## Limitations

Our sample was comprised of a self-selected voluntary panel of vaccine delivery and implementation experts, which may not be representative of all immunization experts. Specifically, individuals with grassroots immunization experience who are instrumental to vaccine program implementation, such as community health workers, were less likely to have been identified through our recruitment strategy. Therefore, our results may not capture the localized challenges experienced by these grassroots individuals. Furthermore, given the global focus of this work, results may not be necessarily applicable at the local level. The survey was also only conducted in English, thereby limiting the participant of those who could engage with the Delphi process. Further, although we collected data on panelists' region of work experience, this may not reflect their nationality or lived experience, limiting our ability to understand

geographic differences amongst experts. Nuanced research is needed to better capture lived experiences in global health research, and weigh this against domain expertise. Finally, our study is limited by the changing COVID-19 situation over time as the Delphi process unfolded (April 2021 to October 2021). This is evident in our results as the identified solutions have different time horizons for impact; some are immediately actionable but of lower potential impact, whereas others are more longer-term but could have a larger, more enduring impact.

## Conclusions

Our study provides a consensus-based ranking of the priority challenges and solutions for COVID-19 vaccine delivery in LMICs. These results can, and should, be used by global health organizations to efficiently and optimally direct resources on these key challenges and solutions. The validity of our findings is strengthened by the use of a Delphi survey, a formal, systematic method of finding consensus among a group of experts. The seniority of the respondents and the international scope of this study also help consolidate our results. While some of the identified challenges were known and predictable, this indicates the need for the global community to be committed to these issues in ongoing pandemic preparedness.

## Supporting information

**S1 Table. Sources for identifying Delphi panelists with vaccine delivery and implementation expertise.**
(DOCX)

**S2 Table. Identification and ranking of important challenges and solutions for the implementation of COVID-19 vaccination programs in low- and middle-income countries.**
(DOCX)

## Acknowledgments

The authors thank the study panelists for their contributions to this study.

## Author Contributions

**Conceptualization:** Archchun Ariyarajah, Isha Berry, Victoria Haldane, Miranda Loutet, Fabio Salamanca-Buentello, Ross E. G. Upshur.

**Data curation:** Archchun Ariyarajah, Isha Berry, Victoria Haldane, Miranda Loutet.

**Formal analysis:** Archchun Ariyarajah, Isha Berry, Victoria Haldane, Miranda Loutet.

**Funding acquisition:** Archchun Ariyarajah, Isha Berry, Victoria Haldane, Miranda Loutet.

**Investigation:** Archchun Ariyarajah, Isha Berry, Victoria Haldane, Miranda Loutet.

**Methodology:** Archchun Ariyarajah, Isha Berry, Victoria Haldane, Miranda Loutet, Fabio Salamanca-Buentello, Ross E. G. Upshur.

**Project administration:** Archchun Ariyarajah, Isha Berry, Victoria Haldane, Miranda Loutet, Fabio Salamanca-Buentello, Ross E. G. Upshur.

**Supervision:** Fabio Salamanca-Buentello, Ross E. G. Upshur.

**Writing – original draft:** Archchun Ariyarajah, Isha Berry, Victoria Haldane, Miranda Loutet.

**Writing – review & editing:** Archchun Ariyarajah, Isha Berry, Victoria Haldane, Miranda Loutet, Fabio Salamanca-Buentello, Ross E. G. Upshur.

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
