## [Decision Letter · Decision Letter 0]

14 May 2022

PGPH-D-22-00475

Identifying Priority Challenges and Solutions for COVID-19 Vaccine Delivery in Low- and Middle-Income Countries: A Modified Delphi Study

Dear Dr. Upshur,

Thank you for submitting your manuscript to PLOS Global Public Health. After careful consideration, we feel that it has merit but does not fully meet PLOS Global Public Health’s publication criteria as it currently stands. Therefore, we invite you to submit a revised version of the manuscript that addresses the points raised during the review process.

Please submit your revised manuscript by . If you will need more time than this to complete your revisions, please reply to this message or contact the journal office at globalpubhealth@plos.org. Please include the following items when submitting your revised manuscript:

We look forward to receiving your revised manuscript.

Kind regards,

Javier H Eslava-Schmalbach, M.D., Ph.D., MSc

Academic Editor

Journal Requirements:

1. Please update your Competing Interests statement. If you have no competing interests to declare, please state: “The authors have declared that no competing interests exist.”

2. In the online submission form, you indicated that “Due to risk of identification, data cannot be made publicly available. However, specific de-identified dataset versions can be made available upon reasonable request to the study team.”. All PLOS journals now require all data underlying the findings described in their manuscript to be freely available to other researchers, either 1. In a public repository, 2. Within the manuscript itself, or 3. Uploaded as supplementary information.

3. Please provide separate figure files in .tif or .eps format only and ensure that all files are under our size limit of 10MB.

Additional Editor Comments (if provided):

Dear Authors: We have finally received comments from our reviewers. Please answer/comment/include each one of their comments

Reviewers' comments:

Reviewer's Responses to Questions

**Comments to the Author**

1. Does this manuscript meet PLOS Global Public Health’s publication criteria? Is the manuscript technically sound, and do the data support the conclusions? The manuscript must describe methodologically and ethically rigorous research with conclusions that are appropriately drawn based on the data presented.

Reviewer #1: Yes

Reviewer #2: Yes

2. Has the statistical analysis been performed appropriately and rigorously?

Reviewer #1: Yes

Reviewer #2: Yes

3. Have the authors made all data underlying the findings in their manuscript fully available (please refer to the Data Availability Statement at the start of the manuscript PDF file)?

Reviewer #1: Yes

Reviewer #2: Yes

4. Is the manuscript presented in an intelligible fashion and written in standard English?

Reviewer #1: Yes

Reviewer #2: Yes

5. Review Comments to the Author

Reviewer #1: Dear authors, I enjoyed reading your article and I think its importance lies in the high level of the respondents who were certified to be 'experts' in implementation of vaccine programs and vaccine delivery. The challenges are highly relatable, having had personal and firsthand experience with some of these challenges. The solutions, I think, are also tenable and practicable as some of them have already been deployed in some places and have been shown to work.

My only reservations were in the recruitment of 'experts' and the response rate, though some of the reasons for these have been well captured in the ‘Limitations’ section of this paper.

However, on a personal note, especially given my experience working in the grassroots in Africa, involving workers in the last mile of the vaccine supply chain as respondents will elucidate more practical and localized challenges and solutions to vaccine delivery and implementation of vaccine programs. This is because they have first hand experience on some of these challenges which, many a time, go unreported to the higher level 'experts'. Also, based on their experience, they may have more local solutions to proffer than the top level 'experts'. This is because, in many cases, they figure out how to navigate some of these challenges on their own without input from or reports to the high level 'experts' for them to take note of.

The only other issue I have is with the ‘Results’ section. I felt submerged into this section without any ‘forewarning’ of how the section will be presented. This made me to step back a couple of times. You might consider including a short passage on how the section will be presented (based on the rounds and the challenges and solutions identified),

Reviewer #2: 1. Recommendation: It'd be beneficial, to come up with plots, depicting the data visually.

2. It'd be nice to mention in detail, how are the panelists chosen for this study.

Did the authors chose people who have worked in developed, developing countries in supply chain management of the vaccines.

6. PLOS authors have the option to publish the peer review history of their article (what does this mean?). If published, this will include your full peer review and any attached files.

**Do you want your identity to be public for this peer review?** For information about this choice, including consent withdrawal, please see our Privacy Policy.

Reviewer #1: **Yes: **Otuto Amarauche Chukwu

Reviewer #2: No

---

## [Editor Report · Decision Letter 1]

27 Jun 2022

PGPH-D-22-00475R1

Identifying Priority Challenges and Solutions for COVID-19 Vaccine Delivery in Low- and Middle-Income Countries: A Modified Delphi Study

Dear Dr. Upshur,

Thank you for submitting your manuscript to PLOS Global Public Health. After careful consideration, we feel that it has merit but does not fully meet PLOS Global Public Health’s publication criteria as it currently stands. Therefore, we invite you to submit a revised version of the manuscript that addresses the points raised during the review process.

Please submit your revised manuscript by . If you will need more time than this to complete your revisions, please reply to this message or contact the journal office at globalpubhealth@plos.org. Please include the following items when submitting your revised manuscript:

We look forward to receiving your revised manuscript.

Kind regards,

Javier H Eslava-Schmalbach, M.D., Ph.D., MSc

Academic Editor

Journal Requirements:

Additional Editor Comments (if provided):

Dear authors

Please resubmit your article, adding the Supplementary Materials that were not included in the current submission.
---

## [Editor Report · Decision Letter 2]

7 Jul 2022

Identifying Priority Challenges and Solutions for COVID-19 Vaccine Delivery in Low- and Middle-Income Countries: A Modified Delphi Study

PGPH-D-22-00475R2

Dear Dr. Upshur,

We are pleased to inform you that your manuscript 'Identifying Priority Challenges and Solutions for COVID-19 Vaccine Delivery in Low- and Middle-Income Countries: A Modified Delphi Study' has been provisionally accepted for publication in PLOS Global Public Health.

Best regards,

Javier H Eslava-Schmalbach, M.D., Ph.D., MSc

Academic Editor